# Molecular Characteristics of Circ_002156 and Its Effects on Proliferation and Differentiation of Caprine Skeletal Muscle Satellite Cells

**DOI:** 10.3390/ijms252312745

**Published:** 2024-11-27

**Authors:** Yuanhua Gu, Jiyuan Shen, Zhiyun Hao, Huimin Zhen, Xinmiao Wu, Jiqing Wang, Mingna Li, Chunyan Ren, Yuan Liu, Yuan Zhao, Pan Yang, Xuanyu Wang

**Affiliations:** Gansu Key Laboratory of Herbivorous Animal Biotechnology, Faculty of Animal Science and Technology, Gansu Agricultural University, Lanzhou 730070, China; guyh202209@163.com (Y.G.); shenjy@st.gsau.edu.cn (J.S.); haozy@gsau.edu.cn (Z.H.); zhenhm@st.gsau.edu.cn (H.Z.); wuxinmiao2020@163.com (X.W.); limn@gsau.edu.cn (M.L.); renyaya86@126.com (C.R.); ly482098@163.com (Y.L.); 18709405439@163.com (Y.Z.); 15709319461@163.com (P.Y.); gsauwxy@163.com (X.W.)

**Keywords:** goat, circ_002156, skeletal muscle satellite cells, proliferation, differentiation

## Abstract

The proliferation and differentiation of skeletal muscle satellite cells (SMSCs) are responsible for the development of skeletal muscle. In our previous study, circ_002156 was found to be highly expressed in caprine *Longissimus Dorsi* muscle, but the regulatory role of the circular RNAs (circRNA) in goat SMSCs remains unclear. In this study, the authenticity of circ_002156 was validated, and its structurally characteristic and cellular localization as well as tissue expression of circ_002156 and its parent genes were investigated. Moreover, the effects of circ_002156 on the viability, proliferation, and differentiation of SMSCs were also studied. The circ_002156 is located on caprine chromosome 19 with a length of 36,478. The circRNA structurally originates from myosin heavy chain 2 (*MYH2*), *MYH1*, and *MYH4* as well as intergenic sequences among the parent genes. RT-PCR and Sanger sequencing confirmed the authenticity of circ_002156. Most circ_002156 (55.5%) was expressed in the nuclei of SMSCs, while 44.5% of circ_002156 was located in the cytoplasm. The circ_002156 and its three parent genes had higher expression levels in the *triceps brachii*, *quadriceps femoris*, and *longissimus dorsi* muscle tissues than in the other five tissues. The expression of circ_002156 and its parent genes *MYH1* and *MYH4* reached the maximum on day 8 of differentiation, while *MYH2* in expression reached the peak on day 4 after differentiation. The Pearson correlation coefficients revealed that circ_002156 had moderate or high positive correlations with the three parent genes in the expression of both *quadriceps femoris* muscle and SMSCs during different differentiation stages. The small interfering RNA circ_002156 (named si-circ_002156) remarkably increased the viability of the SMSCs. The si-circ_002156 also increased the number and parentage of Edu-labeled positive SMSCs as well as the expression levels of four cell proliferation marker genes. These suggest that circ_002156 inhibited the proliferation of SMSCs. Meanwhile, si-circ_002156 decreased the area of MyHC-labeled positive myotubes, the myotube fusion index, and myotube size as well as the expression of its three parent genes and four cell differentiation marker genes, suggesting a positive effect of circ_002156 on the differentiation of SMSCs. This study contributes to a better understanding of the roles of circ_002156 in the proliferation and differentiation of SMSCs.

## 1. Introduction

Circular RNAs (circRNAs) are a class of non-coding RNAs with a closed-circular structure formed by the back-splicing of linear RNAs. The circRNAs were first discovered in plant viroids in the 1970s [1]. Due to their special ring structure, circRNAs had the characteristics of high conservation and stable expression, and are not easily degraded by RNA exonucleases [2,3]. The circRNAs can be divided into three main types according to their origin in the genome including exonic circRNAs (EcRNAs), circular intronic RNAs (CiRNAs), and exon-intron circRNAs (EIciRNAs) [4]. The type and cellular localization of circRNAs are responsible for their functions. For example, EcRNAs mainly located in the cytoplasm can adsorb specific microRNAs (miRNAs), eventually weakening the inhibition of the target mRNAs by miRNAs. EIciRNAs and CiRNAs are mainly distributed in the nucleus, and they primarily regulate the transcription of the parent genes [5]. In addition, the minority of circRNAs play their biological functions by regulating alternative splicing, acting as protein molecular sponges, or encoding small peptides [6].

In recent years, high-throughput RNA-sequencing (RNA-Seq) and in vitro cell experiments have confirmed that circRNAs play important roles in regulating the growth and development of skeletal muscle in animals. For example, circLMO7 was found to increase the expression level of *LMO7* by adsorbing miR-378a-3p, eventually promoting proliferation and inhibiting the differentiation of bovine myoblasts [7]. Li et al. (2018) found that circFGFR4 increased the expression of Wnt family member 3A (*Wnt3a*) by sponging miR-107, and then promoted the differentiation of bovine muscle cells [8]. Ouyang et al. (2018) showed that circSVIL increased the proliferation and differentiation of chicken myoblasts by acting as an miR-203 sponge [9].

While the importance of circRNAs in skeletal muscle activities has been well-studied in cattle and chicken, to our knowledge, few studies have been reported on the effect of a single circRNA on skeletal muscle development in goats. Zhang et al. (2021) found that in in vitro experiments, circ_usp13 significantly promoted the differentiation of goat skeletal muscle satellite cells (SMSCs) and inhibited cell apoptosis [10]. Zhen et al. (2022) showed that circ_003628 promoted the differentiation and proliferation of goat SMSCs [11]. In another study, the interference of circ_UBE3A inhibited the proliferation, viability, and differentiation of goat SMSCs [12].

In our previous study, the expression of circ_002156 was 1.46-fold higher in the *longissimus dorsi* muscle of Liaoning cashmere goats with higher carcass weight and lion-eye area compared with Ziwuling black goats [13]. The result suggests that circ_002156 may be involved in caprine skeletal muscle growth and development. However, the molecular characteristics of circ_002156 and its effect on goat SMSCs remain unclear. According in this study, we investigated the molecular origin of circ_002156 derived from the parent genes, cellular localization of the SMSCs, and tissue expression in nine different caprine tissues. Furthermore, we analyzed the effects of circ_002156 on the viability, proliferation, and differentiation of goat SMSCs. The results provide new insights for elucidating the role of circ_002156 in skeletal muscle growth and development.

## 2. Results

### 2.1. Characteristics and Cellular Localization of Caprine Circ_002156

The comparison of circ_002156 sequences obtained from RNA-Seq data with Caprine Reference Genome Assembly ARS1 indicated that the circRNA is located on the caprine chromosome 19, and is also an exon-intron circRNA with a total length of 37,478 bp. The circ_002156 is derived from caprine myosin heavy chain 2 (*MYH2*), *MYH1,* and *MYH4*. The circ_002156 structurally includes 17 exons (exon 22-exon 38), 15 introns (intron 23-intron 37), and 3′-untranslated regions (UTR) of *MYH2*; 38 exons (exon 2–exon 39) and 3′ UTR of *MYH1*; and also contains 21 exons (exon 1-exon 21), 20 introns (intron 1-intron 20), and 5′ UTR of *MYH4* as well as intergenic sequences among the three parent genes (Figure 1A). The agarose gel electrophoresis results revealed that the Reverse transcription-polymerase chain reaction (RT-PCR) assay produced a band of the expected size for circ_002156 including head-to-tail splicing sequences using divergent primers (Figure 1B). The results from Sanger sequencing revealed that the head-to-tail junction site of circ_002156 was located between *MYH2* and *MYH4*. The result was in accordance with the RNA-Seq results (Figure 1C). These demonstrate the authenticity of circ_002156.

The Reverse transcription-quantitative PCR (RT-qPCR) results showed that the proportion of the expression levels of the reference gene *U6* in the nuclei and cytoplasm of the SMSCs was 82.6% and 17.4%, while in the other reference gene, *GAPDH*, it was 35.3% and 64.7%, respectively. These show that the nuclei and cytoplasm of the SMSCs were separated successfully. It was further found that most circ_002156 (55.5%) was expressed in the nuclei of the SMSCs, while 44.5% of circ_002156 was located in the cytoplasm (Figure 1D).

### 2.2. The Tissue Expression Characteristics of Circ_002156 and Its Parent Genes

The RT-PCR analysis showed that circ_002156 expression was predominantly detected in various types of muscle tissue and heart, but was not detected or only weakly detected in the liver, spleen, kidneys, testis, and lungs (Figure 2A). The RT-qPCR results further revealed that circ_002156 had the highest expression in all muscle tissues, while it had the least expression in the liver, spleen, kidneys, and testis (*p* < 0.05). Meanwhile, the expression level of circ_002156 was higher in the *longissimus dorsi* muscle of Liaoning cashmere goats than in Ziwuling black goats (*p* < 0.05, Figure 2B). The results of the reference gene *GAPDH* from the RT-PCR are shown in Figure 2C.

In general, the results from the RT-PCR and RT-qPCR analyses showed that the parent genes *MYH1* (Figure 3A,E), *MYH2* (Figure 3B,F), and *MYH4* (Figure 3C,G) had higher expression levels in the muscle tissues than in the other tissues. The RT-PCR results of *GAPDH* are shown in Figure 3D. Meanwhile, the expression level of *MYH2* was higher in the *longissimus dorsi* muscle of the Liaoning cashmere goats than Ziwuling black goats (*p* < 0.05), while the expression levels of *MYH1* and *MYH4* were the opposite. It was notable that *MYH1* was not expressed in the caprine heart, liver, and spleen, while the expression of *MYH2* in the heart was not detected (Figure 3A,B).

Pearson correlation coefficient revealed that circ_002156 had high positive correlations (|r| > 0.7) with the parent genes *MYH1* (r = 0.7952, *p* < 0.01), *MYH2* (r = 0.8251, *p* < 0.01), and *MYH4* (r = 0.8874, *p* < 0.01) in the expression levels of *quadriceps femoris* (Figure 4).

### 2.3. The Expression Levels of Circ_002156 and Its Parent Genes in Caprine SMSCs

To further explore the effect of circ_002156 on SMSC differentiation, we detected the expression levels of circ_002156 and its parental genes at different differentiation stages of caprine SMSCs. In summary, the expression of circ_002156 (Figure 5A) and its parent genes, *MYH1* (Figure 5B) and *MYH4* (Figure 5D), wavily increased during the differentiation process and reached the maximum on day 8 of differentiation, while *MYH2* expression reached its peak on day 4 after differentiation (Figure 5C).

Meanwhile, circ_002156 had high positive correlations with its parent genes *MYH2* (r = 0.8527, *p* < 0.0001) and *MYH4* (r = 0.9615, *p* < 0.0001) in expression level at different differentiation stages of caprine SMSCs, while it had a moderate positive correlation with *MYH1* (r = 0.6360, *p* < 0.05) in expression (Figure 6).

### 2.4. Circ_002156 Inhibits Viability of SMSCs

The structure of si-circ_002156 is shown in Figure 7A. When si-circ_002156 was transfected into SMSCs, the siRNA remarkably reduced the expression level of circ_002156 (Figure 7B, *p* < 0.05). This indicates that si-circ_002156 was successfully transfected into caprine SMSCs and could be used for subsequent experiments.

The CCK-8 assay results showed that the downregulation of circ_002156 in expression significantly increased the viability of SMSCs (Figure 7C, *p* < 0.01), suggesting that circ_002156 inhibited the viability of the SMSCs.

### 2.5. Circ_002156 Inhibits Proliferation of SMSCs

The Edu experiment demonstrated that si-circ_002156 increased the number (Figure 8A) and ratio (Figure 8B, *p* < 0.01) of Edu-labeled positive SMSCs when compared with the si-NC group. Meanwhile, the downregulation of circ_002156 in expression also increased the expression levels of cell proliferation marker genes—cyclin dependent kinase 2 (*CDK2*), proliferating cell nuclear antigen (*PCNA*), *CDK4*, and *CyclinE* in SMSCs, but reduced the expression level of *p53* (Figure 8C, *p* < 0.01). These suggest that circ_002156 inhibited the proliferation of SMSCs.

### 2.6. Circ_002156 Promotes Differentiation of SMSCs

When si-circ_002156 was successfully transfected into the SMSCs (Figure 9A; *p* < 0.01), it inhibited the expression levels of the parent genes *MYH1*, *MYH2*, and *MYH4* (*p* < 0.01) (Figure 9B). Moreover, it significantly downregulated the expression levels of the cell differentiation marker genes—myosin heavy chain (*MyHC*), myogenic differentiation (*MyoD*), myogenin (*MyoG*), and myocyte enhancer factor 2C (*MEF2C*) (Figure 9C).

Furthermore, si-circ_002156 also significantly decreased the area of MyHC-labeled positive myotubes (Figure 10A,B; *p* < 0.01) and negatively modulated the number of nuclei in the myotubes (as evaluated by the fusion index) (Figure 10C, *p* < 0.01) and myotube size (nuclei per myotube) (Table 1). These results demonstrate that circ_002156 promoted the differentiation of SMSCs.

## 3. Discussion

In the study, we confirmed the expression of circ_002156 in caprine skeletal muscle tissue and the presence of a head-to-tail splice junction of the circRNA using RT-PCR and DNA sequencing, together displaying the authenticity of circ_002156. Subsequently, the origin, cellular localization, and expression of circ_002156 in the SMSCs during different differentiation periods and its effect on SMSC activities were also reported.

The sequence analysis results revealed that circ_002156 originated from partial sequences of caprine *MYH1*, *MYH2*, and *MYH4* as well as intergenic sequences among the three parent genes. These suggest that circ_002156 belongs to a type of EIciRNA. The finding from the cellular localization analysis that circ_002156 is mainly expressed in the nuclei of SMSCs also supports its classification result, as EIciRNA is mainly distributed in the nucleus. Previous studies have shown that the main role of the type of EIciRNA is to regulate the expression of the parent genes. In this context, the regulatory role of circ_002156 was subsequently confirmed by analyzing the correlation in expression level between circ_002156 and its parent genes.

The RT-qPCR results revealed that circ_002156 exhibited a tissue-specific expression pattern in goats, as it had the highest expression level in various muscles but the lowest expression in the livers. The difference in the expression of circ_002156 across different tissues may be related to its biological function. Meanwhile, circ_002156 had a higher expression level in the *longissimus dorsi* muscle of Liaoning cashmere goats with a higher carcass weight and lion-eye area. These suggest that circ_002156 plays a main role in caprine muscle tissue growth and development. It has been confirmed by previous studies that the biological function of circRNAs is closely related to their parent genes. Circ_002156 is derived from *MYH1*, *MYH2*, and *MYH4*, genes that are all members of the MyHC gene family. Our study demonstrated that the three parent genes were highly expressed in various caprine muscle tissue. MyHC proteins are the basic structural units of myosin, which forms the main component of muscle cells. *MYH1*, *MYH2*, and *MYH4* encode the MyHC-IIx, MyHC-IIa, and MyHC-IIb proteins, respectively. The three proteins are referred to as type II muscle fibers [14]. Type II muscle fibers are closely related to the meat production performance and meat quality of animals [15,16]. Studies have shown that mice with *MYH1* and *MYH4* knocked out had severe skeletal muscle hypoplasia [17]. The Glu706Lys variation in *MYH2* is associated with familial congenital myopathy in humans [18]. In this study, it was concluded that circ_002156 derived from *MYH1*, *MYH2*, and *MYH4* may contribute to caprine skeletal muscle growth and development.

The *quadriceps femoris* muscle of 10 Ziwuling black goats was used to analyze the correlation between circ_002156 and its three parental genes. The Pearson correlation coefficient showed that the expression level of circ_002156 was positively correlated with the three parent genes in both the *quadriceps femoris* muscle and caprine SMSCs during different differentiation stages. This exhibited the close relationship of circ_002156 with the three parent genes in expression.

The RT-qPCR results showed that as differentiation progressed, the expression level of circ_002156 gradually increased (Figure 5A). The trend of change in the expression level of circ_002156 may be related to the status of SMSCs. At the beginning of the differentiation of SMSCs, the activities of the SMSCs are mainly focused on proliferation. However, with the conduct of SMSC differentiation, cell differentiation gradually dominates. Given the promotion effect of circ_002156 on SMSC differentiation, circRNA had the highest expression on day 8 of differentiation. Similarly, due to the inhibition effect of circ_002156 on SMSC proliferation, the circRNA had the lowest expression at the initiation stage of differentiation. In the study, the expression of the parent genes *MYH1*, *MYH2*, and *MYH4* exhibited significant fluctuations during different differentiation periods. Specifically, the expression levels of *MYH1* and *MYH4* reached their peaks on day 8 of differentiation, while *MYH2* had the highest expression on day 4 after differentiation. Brown et al. (2012) also found that *MYH1*, *MYH2*, and *MYH4* were significantly expressed in the later stages of C2C12 cell differentiation and reached the maximum on day 8 of differentiation [19]. The result of *MYH4* in the C2C12 cells was inconsistent with that from SMSCs. This contradiction may be related to the different type of cells. However, this speculation needs to be verified in future study.

It is well-known that muscle growth and weight primarily depend on the number and size of the muscle fibers as well as the proliferative capacity of SMSCs. SMSCs are muscle-derived stem cells that can form myotubes through the proliferation, fusion, and differentiation of the cells. It can therefore be known that the activity, proliferation, and differentiation of SMSCs are responsible for skeletal muscle growth and development [20,21]. In this context, SMSCs have been used as models to investigate skeletal muscle growth and development in animals [22]. Therefore, an investigation into the effect of circ_002156 on the activity, proliferation and differentiation of SMSCs seems particularly important. In this study, it was found that circ_002156 decreased the viability of SMSCs and the number and ratio of Edu-labeled positive SMSCs as well as the expression of *CDK2*, *PCNA*, *CDK4*, and *CyclinE* but increased *p53* expression. The genes described above are marker genes of cell proliferation. *CDK2* and *CDK4* positively regulate cell proliferation by regulating cell cycle progression [23]. *PCNA* is involved in DNA replication and repair, and is used to identify cells in the S phase of the cell cycle when they are in the state of cell proliferation [24]. *CyclinE* can bind and activate *CDK2* and *CDK1* to control DNA replication [25]. However, *p53* inhibits cell proliferation by negatively regulating the expression of *CDK2* [26]. Some circRNAs have been discovered to inhibit cell proliferation by suppressing the expression of the marker genes. For instance, interfered circFAM188B inhibited chicken SMSC proliferation by inhibiting the expression of *CDK2* and *PCNA* [27]. Taken together, these results suggest that circ_002156 inhibits the viability and proliferation of SMSCs.

It has been found that EIciRNAs mainly located in the nuclei can regulate the expression of their parent genes to perform biological functions [5,28]. In this context, we studied the effect of circ_002156 on the expression of its parent genes in the differentiation of SMSCs. A positive regulation effect of circ_002156 on its parent genes *MYH1*, *MYH2*, and *MYH4* was subsequently observed. Given that the MYH gene family has been confirmed to be crucial for the myogenic differentiation process, we therefore speculated that circ_002156 regulates the differentiation of SMSCs via the regulation of *MYH1*, *MYH2*, and *MYH4*.

In this study, circ_002156 significantly increased the expression levels of *MyoD*, *MyoG*, *MyH*C, and *MEF2C*, the area of MyHC-labeled myotubes, the number of nuclei in myotubes, and myotube size. These genes positively regulate myogenic differentiation [29]. As core myogenic regulators, *MyoD1* promotes the transformation of myosatellite cells into myocytes, while *MyoG* controls myoblast fusion [30,31]. The MyHC protein is the basic unit of myosin, and its expression level reflects the fusion ability of myocytes into myotubes during myogenic differentiation [32]. Myocyte enhancer MEF2C is activated by myogenic regulatory factors (MRFs) during the terminal differentiation of myogenic stem cells and collaboratively promotes myoblast fusion and differentiation [33]. Zhuang et al. (2023) found that the knockdown of circKANSL1L downregulated the expression levels of *MyoD*, *MyoG*, and *MyHC*, suggesting that circKANSL1L promotes myoblast differentiation [34]. These results suggest that circ_002156 promotes the differentiation of SMSCs. MyHC is usually used as a myogenic marker protein as the area of myotube marked by MyHC is proportional to the differentiation degree of SMSCs [35]. In the study, circ_002156 increased the area of MyHC-labeled myotubes, the myotube fusion index, and myotube size. It can expedite and enhance the fusion of myoblasts into myotubes, further showing the positive regulatory of circ_002156 in SMSC differentiation.

## 4. Materials and Methods

### 4.1. Sample Collection

All animal experiments were approved by the Animal Experiment Ethics Committee of Gansu Agricultural University (Approval number GSAU-ETH-AST-2021-028).

All experimental animals were sourced from the Yongfeng breeding cooperative (Huanxian Country, Qingyang City, China). Four healthy nine-month-old Ziwuling black goat rams were slaughtered, and nine samples of *longissimus dorsi* muscle, *quadriceps femoris* muscle, *triceps brachii* muscle, heart, liver, spleen, lung, kidney, and testis were collected from each goat. Meanwhile, under the same feeding and management conditions, the *longissimus dorsi* muscle tissues of four healthy nine-month-old Liaoning cashmere goat rams were collected. All tissue samples were washed with DEPC water and stored in liquid nitrogen for RNA extraction.

### 4.2. Cell Culture

According to the methods previously described [36], the SMSCs were isolated, cultured, and finally frozen in liquid nitrogen for preservation. In this study, frozen SMSCs were placed in a 37 °C water bath and shook quickly to resuscitate the cells. After centrifugation at 1200 rpm for 5 min, the supernatant was discarded and the remaining cells were resuspended using the growth medium containing a DMEM/F12 medium (Hyclone, Logan, UT, USA) supplemented with 20% fetal bovine serum (Invigentech, Carlsbad, CA, USA), 10% horse serum (Invigentech, Carlsbad, CA, USA), and 1% penicillin/streptomycin (Hyclone, Logan, UT, USA). The SMSCs were finally cultured in an incubator at 37 °C with 5% CO_2_.

### 4.3. The Authenticity Validation of Circ_002156

The RT-PCR and Sanger sequencing were used to verify the authenticity of circ_002156. Divergent primers were designed to amplify the sequences of circ_002156, which included the head-to-tail splicing junction site (Table 1). The RT-PCR was performed using the Applied Biosystems SimpliAmp Thermal Cycler (Thermo Fisher Scientific, Waltham, MA, USA). The amplicon was subsequently visualized and sequenced using 1.5% agarose gel electrophoresis (Servicebio, Wuhan, China) and DNA sequencing, respectively. The sequences from DNA sequencing were aligned to sequences for RNA-Seq and Caprine Genome Assembly ARS1 sequences using MEGA 7.0 software (Institute for Genomics and Evolutionary Medicine, Temple University, Philadelphia, PA, USA). The results can validate the presence of the backing-splicing junction site of circ_002156, further confirming the authenticity of the circRNA.

### 4.4. Cellular Localization of Circ_002156 in SMSCs

The Minute^TM^ Nucleoplasmic Isolation Kit (Invent, Minneapolis, MN, USA) was utilized to isolate the nuclei and cytoplasm of the SMSCs. Subsequently, RNA extracted from the SMSCs was reverse transcribed to generate cDNA. The RT-qPCR was employed to detect of the expression levels of circ_002156 in the cytoplasm and nuclei of the SMSCs. The two internal reference genes *U6* and *β-actin* were selected to standardize the expression levels in the nuclei and cytoplasm, respectively [36]. The primers used for amplifying the sequences are provided in Table 2.

### 4.5. RNA Isolation and RT-qPCR

The total RNA from the nine caprine tissues and SMSCs during different differentiation periods was extracted using TRIzol reagent (Invitrogen, Carlsbad, CA, USA). To check the integrity of the RNA, 1.5% agarose gel electrophoresis was used, and a NanoDrop 8000 spectrophotometer (NanoDrop Technologies, Wilmington, NC, USA) was used to detect the concentration and purity of the RNA. A HiScript III 1st Strand cDNA Synthesis Kit (Vazyme, Nanjing, China) was used to synthesize the cDNA. Table 1 shows the designed primers.

RT-qPCR was performed in triplicate in a 20-μL reaction including 10 μL SYBR qPCR Master Mix, 2 μL cDNA, 0.8 μL primers, and 7.2 μL RNase-free ddH_2_O. The thermal profile was 95 °C for 10 min, followed by 40 cycles of 95 °C for 30 s, 60 °C for 30 s, and 72 °C for 30 s. *GAPDH* was used as an internal control and the results were statistically analyzed using the 2^−ΔΔCT^ method.

### 4.6. Transfection of Circ_002156 into SMSCs

The small interfering RNA of circ_002156 (named si-circ_002156) and its negative controls (named si-NC) were synthesized by GenePharma (GenePharma, Shanghai, China). When the density of the SMSCs reached approximately 70–80%, si-circ_002156 and si-NC were transfected into the cells using the INVI DNA and RNA Transfection Reagent^TM^ (Invigentech Carlsbad, CA, USA). The expression level of circ_002156 was detected by RT-qPCR to determine the transfection efficiency.

### 4.7. Viability and Proliferation of SMSCs Detected

The viability and proliferation of the SMSCs were detected with the cell counting kit-8 (CCK-8) and 5-ethynyl-20-deoxyuridine (Edu) assays, respectively. When the density of the SMSCs cultured in 96-well plates reached 70% to 80%, si-circ_002156 and si-NC were transfected into the SMSCs, respectively. After transfection for 48 h, a total of 100 μL 10% CCK-8 reagent (Vazyme, Nanjing, China) was added to each well and then cultured for 2 h in a 37 °C incubator. The absorbance of SMSCs at 450 nm was detected using a microplate reader (Thermo Scientific, Waltham, MA, USA).

After the transfection of si-circ_002156 and si-NC for 44 h, 20 μM of Edu reagent (Vazyme, Nanjing, China) was added to each well and then cultured for 4 h, and the number of Edu-labeled positive cells was detected using an IX73 (Olympus, Tokyo, Japan) microscope. Three images were randomly captured for each transfected group, and the percentage of Edu-labeled positive SMSCs was calculated using ImageJ 1.53k software (National Institutes of Health, Bethesda, Montgomery, MD, USA).

Meanwhile, to further validate the effect of circ_002156 on the proliferation of SMSCs, the total RNA of the SMSCs was extracted 48 h after transfection, and the expression levels of five cell proliferation marker genes *CDK2*, *CDK4*, *PCNA*, *CyclinE*, and *p53* were detected using RT-qPCR. *β-actin* was used as an internal control, and its primer sequence information is shown in Table 1.

### 4.8. Effect of Circ_002156 on Differentiation of SMSCs

When the confluence of SMSCs reached approximately 80%, the induced differentiation of the SMSCs was initiated by replacing the growth medium with differentiation medium (97% DMEM/F12 medium, 2% horse serum, and 1% penicillin/streptomycin). The total RNA of the SMSCs was extracted on days 0, 2, 4, 6, and 8 after differentiation, and the expression levels of circ_002156 and its parent genes were detected using RT-qPCR.

When the confluence of SMSCs achieved 80% to 90%, si-circ_002156 and si-NC were transfected into the SMSCs based on the method described above. After transfection for 24 h, the SMSCs were induced to differentiate using the differentiation medium. On day 6 after differentiation, MyHC immunofluorescence staining was used to identify myotubes of the SMSCs. Briefly, the SMSCs were fixed for 30 min in 200 μL of 4% paraformaldehyde (Servicebio, Wuhan, China) and blocked for 30 min using 5% bovine serum albumin (BSA, Servicebio, Wuhan, China). The primary antibody monoclonal anti-MyHC antibody (R&D Systems, Minneapolis, MN, USA) was used to stain the SMSCs overnight at 4 °C. The cells were washed 3 times using PBS. Then, the corresponding secondary antibody goat anti-mouse IgG (H+L) cy3-conjugated (Affinity, Melbourne, Australia) was used to incubate the SMSCs for 2 h at room temperature. The nuclei of the SMSCs were visualized and observed using the Hoechst 33342 (Solarbio, Wuhan, China) stain and an IX73 microscope (Olympus, Tokyo, Japan), respectively. Meanwhile, on day 6 of SMSC differentiation, the total RNA from the cells transfected with si-circ_002156 and si-NC was extracted, and RT-qPCR was used to detect the expression levels of circ_002156, its parent genes *MYH1*, *MYH2,* and *MYH4*, and the myogenic differentiation marker genes *MyHC*, *MyoD*, *MyoG*, and myocyte enhancer factor 2C (*MEF2C*) of the SMSCs. *β-tubulin* was used as an internal reference (Table 1).

### 4.9. Statistical Analysis

All experimental data were analyzed and expressed as the means ± standard deviation (SD) using SPSS 22.0 software (IBM, Armonk, NY, USA). The difference between the two groups was analyzed using the two-tailed student’s *t*-test, while the difference among multiple groups was compared using one-way ANOVA.

## 5. Conclusions

Our results describe the molecular characteristics of circ_002156 regarding its origin sequences, cellular localization, and specific expression in various tissues and SMSCs during different differentiation periods. Moreover, circ_002156 was found to inhibit the proliferation of goat SMSCs but promote the differentiation of the cells. This study lays the foundation for a better understanding of the biological function of circ_002156 in caprine skeletal muscle growth and development.

## Figures and Tables

**Figure 1 ijms-25-12745-f001:**
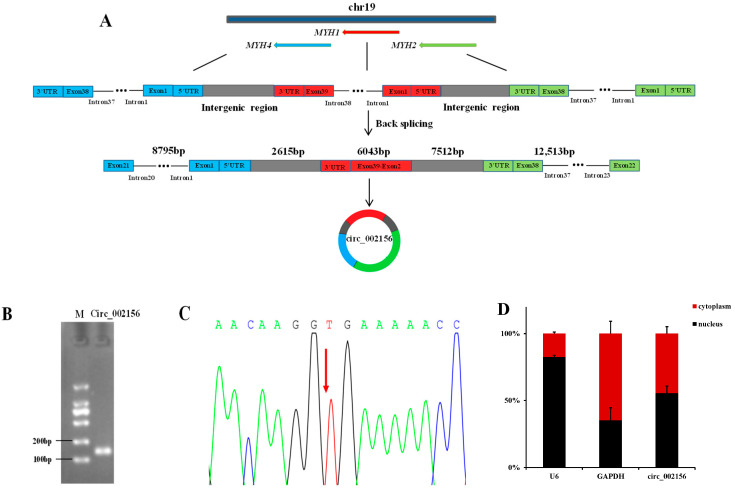
Structure, identification, and cellular localization of circ_002156. (**A**) Structure diagram of circ_002156 derived from the parent genes myosin heavy chain 1(*MYH1*), *MYH2*, and *MYH4*. (**B**,**C**) The authenticity validation of circ_002156 using the RT-PCR assay and Sanger sequencing. (**D**) The cellular localization of circ_002156 in the goat skeletal muscle satellite cells (SMSCs) with *GAPDH* and *U6* being the reference genes. M—marker. The head-to-tail splice junction site of circ_002156 was labeled using a red arrow.

**Figure 2 ijms-25-12745-f002:**
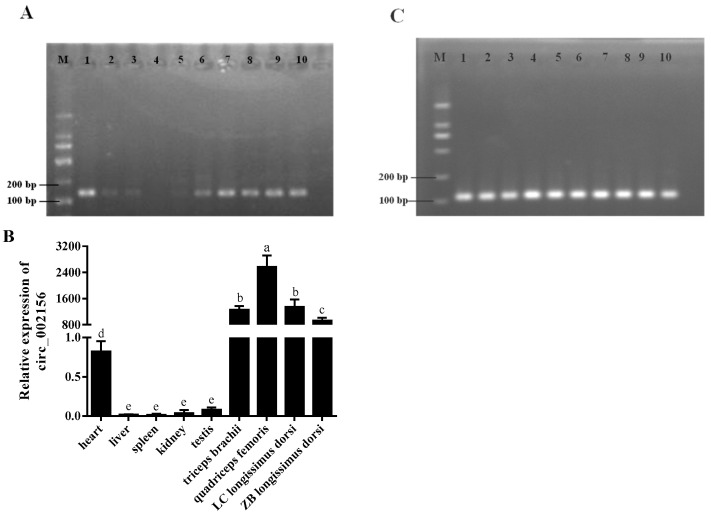
The expression of circ_002156 (**A**,**B**) and *GAPDH* (**C**) in different tissues of goats detected using RT-PCR and RT-qPCR, respectively. M—marker; 1—heart; 2—liver; 3—spleen; 4—lungs; 5—kidneys; 6—testis; 7—*triceps brachii*; 8—*quadriceps femoris*; 9—*longissimus dorsi* of Liaoning cashmere goat (LC); 10—*longissimus dorsi* of Ziwuling black goat (ZB). Since circ_002156 was not expressed in the lung, it was not included in Figure 2. The different lowercase letters indicate significant differences (*p* < 0.05).

**Figure 3 ijms-25-12745-f003:**
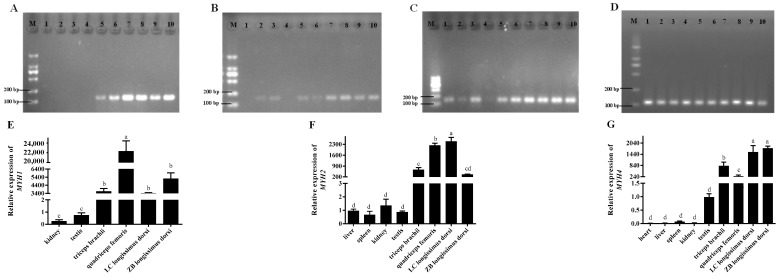
The expression of the parent genes *MYH1* (**A**,**E**), *MYH2* (**B**,**F**), *MYH4* (**C**,**G**), and *GAPDH* (**D**) in different caprine tissues detected using RT-PCR and RT-qPCR, respectively. M—marker; 1—heart; 2—liver; 3—spleen; 4—lungs; 5—kidneys; 6—testis; 7: *triceps brachii*; 8—*quadriceps femoris*; 9—*longissimus dorsi* of Liaoning cashmere goat (LC); 10—*longissimus dorsi* of Ziwuling black goat (ZB). The different lowercase letters indicate significant differences (*p* < 0.05).

**Figure 4 ijms-25-12745-f004:**
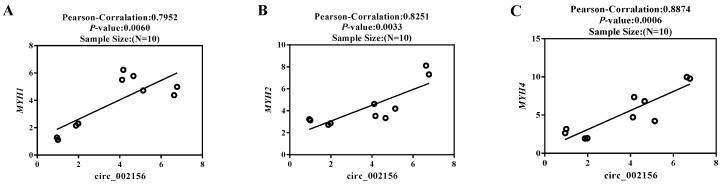
Pearson correlation analysis in the expression of *quadriceps femoris* between circ_002156 and the parent genes *MYH1* (**A**), *MYH2* (**B**), and *MYH4* (**C**). Each circle represents the expression of the parent gene in different ZiwuLing black goats.

**Figure 5 ijms-25-12745-f005:**
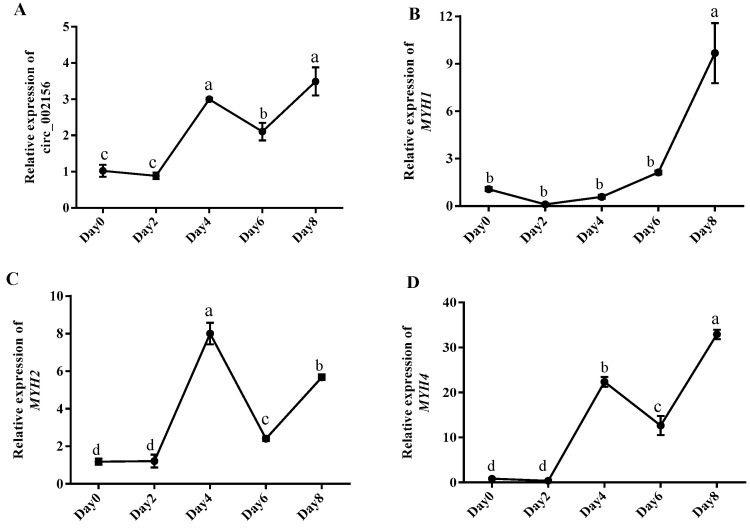
The expression levels of circ_002156 (**A**) and its parent genes *MYH1* (**B**), *MYH2* (**C**), and *MYH4* (**D**) on days 0, 2, 4, 6, and 8 after skeletal muscle satellite cell (SMSC) differentiation. Different lowercase letters represent significant differences (*p* < 0.05).

**Figure 6 ijms-25-12745-f006:**
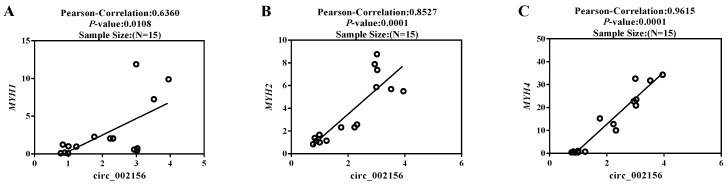
Pearson correlation analysis between circ_002156 and its parent genes *MYH1* (**A**), *MYH2* (**B**), and *MYH4* (**C**) in expression level at different differentiation stages of caprine skeletal muscle satellite cells (SMSCs). Each circle represents the expression of the parent gene in different ZiwuLing black goats.

**Figure 7 ijms-25-12745-f007:**
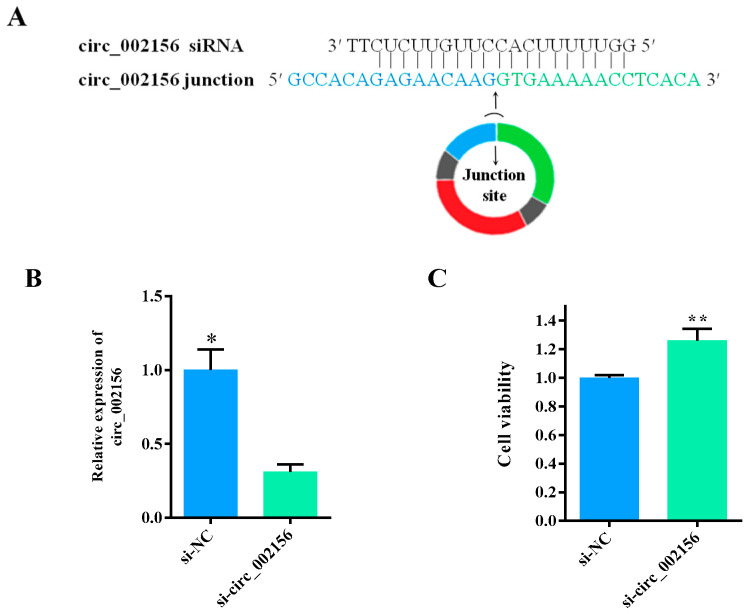
Effect of si-circ_002156 on the viability of the caprine skeletal muscle satellite cells (SMSCs). (**A**) Schematic diagram of the si-circ_002156 sequence completely complementarily binding with the junction site sequences of circ_002156. (**B**) The expression level of circ_002156 detected in the SMSCs transfected with si-circ_002156. (**C**). Effect of si-circ_002156 on the viability of SMSCs investigated using the CCK-8 assay. * *p* < 0.05 and ** *p* < 0.01.

**Figure 8 ijms-25-12745-f008:**
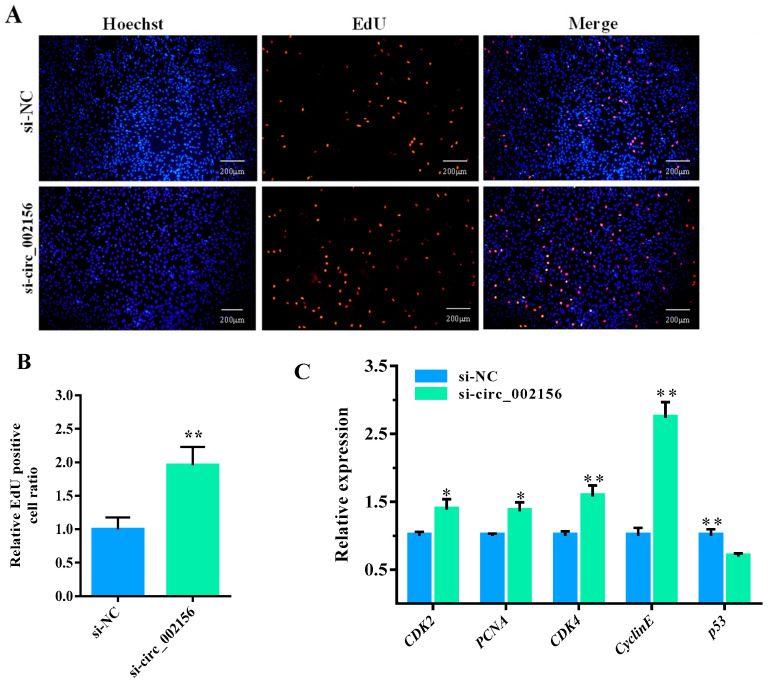
Effect of si-circ_002156 on the proliferation of skeletal muscle satellite cells (SMSCs). (**A**) The proliferation of SMSCs detected using an Edu assay. (**B**) The ratio of Edu-labeled positive SMSCs. (**C**) The expression levels of cell proliferation marker genes—cyclin dependent kinase 2 (*CDK2*), proliferating cell nuclear antigen (*PCNA*), *CDK4*, *CyclinE*, and *p53* detected in SMSCs transfected with si-circ_002156. * *p* < 0.05 and ** *p* < 0.01.

**Figure 9 ijms-25-12745-f009:**
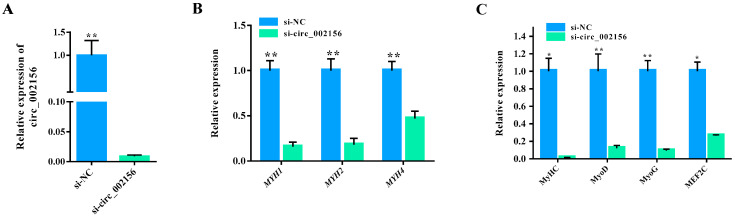
Effect of circ_002156 on the differentiation of goat skeletal muscle satellite cells (SMSCs). (**A**) Transfection efficiency of si-circ_002156 detected. (**B**) The relative expression levels of its three parent genes and (**C**) four cell differentiation marker genes—myosin heavy chain (*MyHC*), myogenic differentiation (*MyoD*), myogenin (*MyoG*), and myocyte enhancer factor 2C (*MEF2C*) in SMSCs transfected with si-circ_002156. * *p* < 0.05 and ** *p* < 0.01.

**Figure 10 ijms-25-12745-f010:**
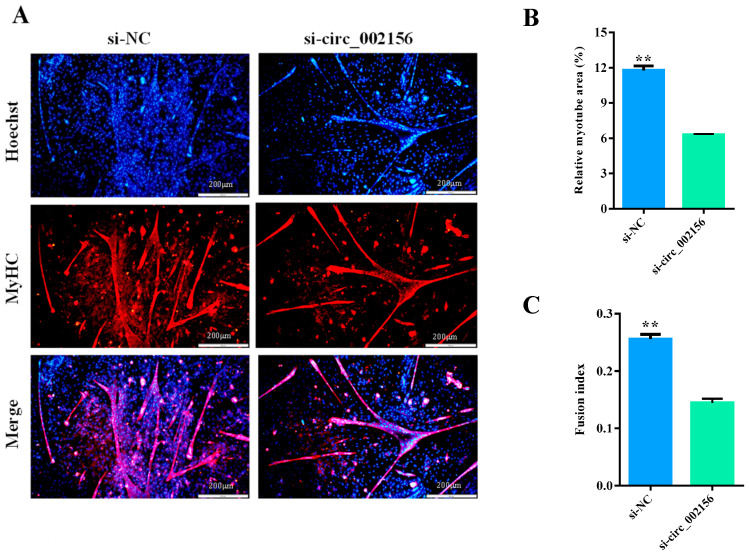
Effect of circ_002156 on myotubes. (**A**) The immunofluorescence results of SMSCs when si-circ_002156 was transfected. (**B**) The area of MyHC-labeled positive myotubes. (**C**) Circ_002156 positively modulated the number of nuclei in the myotubes. ** *p* < 0.01.

**Table 1 ijms-25-12745-t001:** Effect of circ_002156 on nuclei per myotube.

circRNA	Nuclei/Myotube
si-NC	45.976 ± 1.100 **
si-circ_002156	28.562 ± 2.276

** *p* < 0.01.

**Table 2 ijms-25-12745-t002:** Primer sequence information used for RT-qPCR and the siRNA sequences of circ_002156.

Name	Forward (5′→3′)	Reverse (5′→3′)
circ_002156	F: CTGAACTGACGGCCAAGAAG	R: TCATCCAGACCTGCCATCTC
*MYH2*	F: CTGGCTGGAGAAGAACAAGG	R: CACCGTCTGGAAAGAAGAGC
*MYH1*	F: ATGCGGGAACAGTGGACTAC	R: GCACCAGAGAACAGGAAAGC
*MYH4*	F: TCAAGGGGAGATCACAGTCC	R: GAGGTAGGCAGCCTTGTCAG
*CDK2*	F: GACCAGCTCTTCCGGATCTT	R: ACAAGCTCCGTCCATCTTCA
*CDK4*	F: ACTTTGTGGCCCTCAAGAGT	R: CCTGAGGTCTTGGTCCACAT
*PCNA*	F: GAACCTCACCAGCATGTCCAA	R: TTCACCAGAAGGCATCTTTACT
*CyclinE*	F: CTCCCTGATTCCCACACCTG	R: CATAAGATGCTTGTCCCTCA
*p53*	F: CGGCTTGCAGAAACCTCTTT	R: CCCTTTTCTACCTCCTGCCA
*MYHC*	F: CCACATCTTCTCCATCTCTG	R: GGTTCCTCCTTCTTCTTCTC
*MYOD*	F: GTGCAAACGCAAGACGACTA	R: GCTGGTTTGGGTTGCTAGAC
*MYOG*	F: CGTGGGCGTGTAAGGTGT	R: GGCGCTCTATGTACTGGATGG
*MEF2C*	F: ATCCTGATGCAGACGATTCAG	R: GGTGGAACAGCACACAATCTT
*β-tubulin*	F: AGCGTATCTCAGAGCAGTTC	R: AATCCTCTTCCTCTTCTGCG
*GAPDH*	F: ACACTGAGGACCAGGTTGTG	R: GACAAAGTGGTCGTTGAGGG
*β-actin*	F: AGCCTTCCTTCCTGGGCATGGA	R: GGACAGCACCGTGTTGGCGTAA
*U6*	F: GGAACGATACAGAGAAGATTAGC	R: TGGAACGCTTCACGAATTTGCG
si-circ_002156	F: GAGAACAAGGUGAAAAACCTT	R: TGGAACGCTTCACGAATTTGCG
si-NC	F: UUCUCCGAACGUGUCACGUTT	R: GGUUUUUCACCUUGUUCUCTT

## Data Availability

The data presented in this study are available in the article.

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
