# Peer review of "Molecular Characteristics of Circ_002156 and Its Effects on Proliferation and Differentiation of Caprine Skeletal Muscle Satellite Cells"

_ijms, 2024, doi:10.3390/ijms252312745_

Round 1
Reviewer 1 Report
Comments and Suggestions for Authors
This study selected a candidate circRNA (circ_002156) affecting skeletal muscle growth and development based their previous study, and validated the authenticity and performed expression patterns and analyzed the effect on proliferation and differentiation of SMSCs in goat. Following issues need to be considered.
1. Page 6, There is no significance to perform the correlation using the expression of circ_002156 in different tissues. As shown in Figure 4, the large expression differences of circ_002156 were existed among the tissues, so it is naturally to obtained the high correlation using the expression of circ_002156 in different tissues. This fit is like using 2 points to establish a linear regression relationship, which is not reasonable. The authors should establish the linear regression relationship using the expression of circ_002156 in the same tissue.
2. Figure 6B. If MYH1 is the parent gene of circ_002156, why their expression patterns are not consistent?
3. The authors perform the RNAi experiment, but the overexpression experiment is lacking. Moreover, lacking WB experiments to support the conclusion.
4. MYH1, MYH2 and MYH4 genes correspond to MyHC proteins, as the differentiation of SMSCs, their expressions are upregulated. According to the conclusion of this study, circ_002156 can promote the differentiation of SMSCs, it means that circ_002156 may promote the synthesis of MyHC or the expression of MYH1, MYH2 and MYH4 genes. if circ_002156 derived from the MYH1, MYH2 and MYH4 genes, suggesting the loop regulation may be existed among circ_002156 and their parent genes, MYH1, MYH2 and MYH4. So, I suggest the authors explore the relationship circ_002156 and their parent genes using the RNAi and overexpression experiments and the specific regulatory mechanism need to be explored.
Totally, the regulatory mechanism is not clear based the present data.
Author Response
Reviewer: 1
Comments 1:[Page 6, There is no significance to perform the correlation using the expression of circ_002156 in different tissues. As shown in Figure 4, the large expression differences of circ_002156 were existed among the tissues, so it is naturally to obtained the high correlation using the expression of circ_002156 in different tissues. This fit is like using 2 points to establish a linear regression relationship, which is not reasonable. The authors should establish the linear regression relationship using the expression of circ_002156 in the same tissue.]
Response 1: Thank you for your valuable comments and suggestions! We have deleted the correlation analysis of circ_002156 expression in different tissues.
Comments 2:[Figure 6B. If MYH1 is the parent gene of circ_002156, why their expression patterns are not consistent?]
Response 2: It is normal that the expression patterns of parental genes and circRNA are inconsistent. First, they function differently, even within the same organization. circRNA acts mainly by regulating downstream target genes or its parent genes. Parental genes work by directly coding for proteins. On the other hand, several studies have shown that even in the same tissue, there are significant differences in the expression abundance of circRNA and parental genes, most of the parental genes have higher expression than circRNA, while a few show the opposite trend (Wang et al., 2021). Therefore, there is no clear correlation between circRNA and parental gene expression patterns.
Reference
Wang. J., Zhou. H., Hickford. JGH., Hao. Z., Gong. H., Hu. J., Liu. X., Li. S., Shen. J., Ke. N., Song. Y., Qiao. L., Luo. Y. Identification and characterization of circular RNAs in mammary gland tissue from sheep at peak lactation and during the nonlactating period. J Dairy Sci. 2021,104, 2396-2409.。
Comments 3:[ The authors perform the RNAi experiment, but the overexpression experiment is lacking. Moreover, lacking WB experiments to support the conclusion.]
Response 3: Thank you for your valuable comments and suggestions! Since the total length of circ_002156 in this study exceeded 30,000 bp, plasmid construction could not be performed, so no overexpression experiment was conducted. However, our same experimental results were all verified by at least two or more experimental methods. For example, in the cell proliferation experiment, we used CCK8, EDU and qPCR, and the conclusions were all the same. Therefore, only knockdown experiments can also explain the function of the circRNA.
Regarding the lack of WB experiments, on the one hand, we lack primary antibodies specifically for goats, so WB experiments cannot be conducted. On the other hand, for differentiation marker genes, we did immunofluorescence staining of myotubes, which is also quantitative for proteins.
Comments 4:[MYH1, MYH2 and MYH4 genes correspond to MyHC proteins, as the differentiation of SMSCs, their expressions are upregulated. According to the conclusion of this study, circ_002156 can promote the differentiation of SMSCs, it means that circ_002156 may promote the synthesis of MyHC or the expression of MYH1, MYH2 and MYH4 genes. if circ_002156 derived from the MYH1, MYH2 and MYH4 genes, suggesting the loop regulation may be existed among circ_002156 and their parent genes, MYH1, MYH2 and MYH4. So, I suggest the authors explore the relationship circ_002156 and their parent genes using the RNAi and overexpression experiments and the specific regulatory mechanism need to be explored.]
Response 4: Thank you for your valuable comments and suggestions! It has been found that exon-intron circRNAs mainly located in nuclei can regulate the expression of their parent genes to perform biological functions (Hsiao et al., 2017; Li et al., 2015). In this context, we haved added the effect of circ_002156 on the expression of its parent genes, but the mechanism verification experiments are not supplemented. As the reviewer said, and a positive regulation effect of circ_002156 on its parent genes MYH1, MYH2 and MYH4 was subsequently observed (as shown in Figure 9). MYH gene family was confirmed to be crucial for the myogenic differentiation process. We therefore speculated that circ_002156 regulate differentiation of SMSCs via regulation of MYH1, MYH2 and MYH4. However, the detailed molecular mechanism will be verified in future studies.
Reference
Hsiao, K.Y.; Sun, H.S.; Tsai, S.J. Circular RNA-new member of noncoding RNA with novel functions. Exp. Biol. Med. 2017, 242, 1136-1141.
Li, Z.Y.; Huang, C.; Bao, C.; Chen, L.; Lin, M.; Wang, X.L.; Zhong, G.L.; Yu, B.; Hu, W.C.; Dai, L.M.; Zhu, P.F.; Chang, Z.X.; Wu, Q.F.; Zhao, Y.; Jia, Y.; Xu, P.; Liu, H.J.; Shan, G. Exon-intron circular RNAs regulate transcription in the nucleus. Nat. Struct. Mol. Biol. 2015, 22, 256-264.]
Reviewer 2 Report
Comments and Suggestions for Authors
GU et al submit an original research article entitled "Molecular characteristics of circ_002156 and its effects on proliferation and differentiation of caprine skeletal muscle satellite cells". They studied in an extensive way the roles of circ_002156 in proliferation and differentiation of caprine SMSCs. The article is interesting even if the techniques are somewhat dated (use of gels for PCR experiments). Can the authors justify the use of U6 in their experiments as it is smaller than circ_0021561,
Have the authors looked in their slides if circ_002156 is able to modulate the number of nuclei in myotubes and myotube size? See PMID: 8377947 for reference.
As the authors used black goat rams, can they justify the use of the male sex?
Author Response
Reviewer: 2
Comments 1:[GU et al submit an original research article entitled "Molecular characteristics of circ_002156 and its effects on proliferation and differentiation of caprine skeletal muscle satellite cells". They studied in an extensive way the roles of circ_002156 in proliferation and differentiation of caprine SMSCs. The article is interesting even if the techniques are somewhat dated (use of gels for PCR experiments). Can the authors justify the use of U6 in their experiments as it is smaller than circ_002156?]
Response 1: Thank you for your valuable comments and suggestions! First of all, the selection criteria of internal reference genes is based on its expression stability, which is not directly related to its sequence length. A large number of studies have proved that the expression of U6 is stable in different cell types and different treatments, so it is widely used as an internal reference in circRNA cell localization experiments (Pan et al., 2022; Yu et al., 2022; Zhen et al., 2022).
Reference
Pan, Z., Zhao, R., Li, B., Qi, Y., Qiu, W., Guo, Q., Zhang, S., Zhao, S., Xu, H., Li, M., Gao, Z., Fan, Y., Xu, J., Wang, H., Wang, S., Qiu, J., Wang, Q., Guo, X., Deng, L., Zhang, P., Xue, H., Li, G. EWSR1-induced circNEIL3 promotes glioma progression and exosome-mediated macrophage immunosuppressive polarization via stabilizing IGF2BP3. Mol Cancer. 2022, 21, 16.
Yu, L., Zhu, H., Wang, Z., Huang, J., Zhu, Y., Fan, G., Wang, Y., Chen, X., Zhou, G. Circular RNA circFIRRE drives osteosarcoma progression and metastasis through tumorigenic-angiogenic coupling. Mol Cancer. 2022, 21, 167.
Zhen, H.; Shen, J.; Wang, J.; Luo, Y.; Hu, J.; Liu, X.; Li, S.; Hao, Z.; Li, M.; Shi, B.; Gu, Y. Characteristics and Expression of circ_003628 and Its Promoted Effect on Proliferation and Differentiation of Skeletal Muscle Satellite Cells in Goats. Animals (Basel). 2022, 12, 2524.
Comments 2:[Have the authors looked in their slides if circ_002156 is able to modulate the number of nuclei in myotubes and myotube size? See PMID: 8377947 for reference.]
Response 2: Thank you for your valuable comments and suggestions! We have added this part of the data, as shown in Figure 10 and Table 1 of the original article.
Comments 3:[As the authors used black goat rams, can they justify the use of the male sex?]
Response 3: Thank you for your valuable comments and suggestions! The reason for using RAMS in this study is to maintain the scientific and rigorous nature of the study. The circRNA in this study was identified from the previous high-throughput sequencing results of our research group, and the high-throughput sequencing was for Ziwuling black goat RAMS. In order to avoid the interference of sex on the experimental results, male goats were selected as the cell source material in this study.
Reference
Shen, J.Y.; Zhen, H.M.; Li, L.; Zhang, Y.T.; Wang, J.Q.; Hu, J.; Liu, X.; Li, S.B.; Hao, Z.Y.; Li, M.N.; Zhao, Z.D.; Luo, Y.Z. Identification and characterization of circular RNAs in Longissimus dorsi muscle tissue from two goat breeds using RNA-Seq. Mol. Genet. Genomics 2022, 297, 817-831.
Round 2
Reviewer 1 Report
Comments and Suggestions for Authors
The authors have made a revision to the manuscript, but some issues still need to be elucidated.
1、Although the author deleted the correlation analysis of circ_002156 expression in different tissues,I strongly suggest the authors provide the correlation between the expression of circ_002156 and parent genes using the same type of muscle tissues derived from a certain number of individuals.
2、Figure 3 and Figure 4, the results of RT-PCR without reference genes are unacceptable.
3、At present, almost all the figures and tables provided are not satisfactory, it is strongly recommended to rearrange the layout of the figures and improve the resolution.
Comments on the Quality of English Language
1、Language still needs to be improved, and suggest the authors invite an English native speaker to revise the manuscript.
Author Response
Dear Editors and Reviewers:
Thank you for your letter and for the reviewers’ comments concerning our manuscript entitled “Molecular characteristics of circ_002156 and its effects on proliferation and differentiation of caprine skeletal muscle satellite cells” (ID: IJMS-3277584). Those comments are all valuable and very helpful for revising and improving our paper, as well as the important guiding significance to our researches. We have studied comments carefully and have made correction which we hope meet with approval.
Reviewer: 1
Comments 1: [ Although the author deleted the correlation analysis of circ_002156 expression in different tissues, I strongly suggest the authors provide the correlation between the expression of circ_002156 and parent genes using the same type of muscle tissues derived from a certain number of individuals.]
Response 1: Thank you for your valuable comments and suggestions! We selected the same tissue quadriceps femoris from 10 different Ziwuling black goats to analyze the correlation in expression levels between circ_002156 and its three parent genes, as shown in Figure 5 of the original manuscript.
Comments 2: [ Figure 3 and Figure 4, the results of RT-PCR without reference genes are unacceptable.]
Response 2: Thank you for your valuable comments and suggestions! We have added the gel electrophoresis diagram of the internal reference gene GAPDH, as shown in Figure 3 and Figure 4 of the original manuscript.
Comments 3: [ At present, almost all the figures and tables provided are not satisfactory, it is strongly recommended to rearrange the layout of the figures and improve the resolution.]
Response 3: Thank you for your valuable comments and suggestions! We have rearranged some of the figures and improved the resolution of the images.
Reviewer: 2
Comments : [changes are ok.]
Response : Thank you for your guidance and recognition!

Reviewer 2 Report
Comments and Suggestions for Authors
changes are ok
Author Response
Dear Editors and Reviewers:
Thank you for your letter and for the reviewers’ comments concerning our manuscript entitled “Molecular characteristics of circ_002156 and its effects on proliferation and differentiation of caprine skeletal muscle satellite cells” (ID: IJMS-3277584). Those comments are all valuable and very helpful for revising and improving our paper, as well as the important guiding significance to our researches. We have studied comments carefully and have made correction which we hope meet with approval.
Reviewer: 1
Comments 1: [ Although the author deleted the correlation analysis of circ_002156 expression in different tissues, I strongly suggest the authors provide the correlation between the expression of circ_002156 and parent genes using the same type of muscle tissues derived from a certain number of individuals.]
Response 1: Thank you for your valuable comments and suggestions! We selected the same tissue quadriceps femoris from 10 different Ziwuling black goats to analyze the correlation in expression levels between circ_002156 and its three parent genes, as shown in Figure 5 of the original manuscript.
Comments 2: [ Figure 3 and Figure 4, the results of RT-PCR without reference genes are unacceptable.]
Response 2: Thank you for your valuable comments and suggestions! We have added the gel electrophoresis diagram of the internal reference gene GAPDH, as shown in Figure 3 and Figure 4 of the original manuscript.
Comments 3: [ At present, almost all the figures and tables provided are not satisfactory, it is strongly recommended to rearrange the layout of the figures and improve the resolution.]
回应 3:感谢您的宝贵意见和建议!我们重新排列了一些数字,并提高了图像的分辨率。
审稿人:2
评论 : [更改还可以。
回应 : 感谢您的指导和认可!

Round 3
Reviewer 1 Report
Comments and Suggestions for Authors
The authors have revised the manuscript according the suggestions. However, the graph size and layout of the whole paper are still unsatisfactory. If these issues are solved, the paper can be accepted for publication.
Author Response
Comments 1: [ The authors have revised the manuscript according the suggestions. However, the graph size and layout of the whole paper are still unsatisfactory. If these issues are solved, the paper can be accepted for publication.]
Response 1: Thank you for your valuable comments and suggestions! We have re-modified and typed the size and layout of the graphics in the paper, and we also hope to get your guidance.
